# Debiased and Denoised Representation Learning for Incomplete Multi-view Clustering

**Qianqian Wang**[1], **Xurui Liao**[1,*] **Wei Feng**[2], **Quanxue Gao**[1]
[1]Xidian University, [2]Northwest A&F University
`qqwang@xidian.edu.cn, 21159100024@stu.xidian.edu.cn`
`weifeng.ft@foxmail.com, qxgao@xidian.edu.cn`

## Abstract

Multi-view clustering achieves outstanding performance but relies on the assumption of complete multi-view samples. However, certain views may be partially unavailable due to failures during acquisition or storage, resulting in distribution shifts across views. Although some incomplete multi-view clustering (IMVC) methods have been developed based on data imputation, these methods increase the unnecessary computational complexity. Therefore, some methods introduce consensus representation imputation and receives remarkable popularity. Nevertheless, they generally ignore the inter-view distribution bias due to missing views, which further leads semantic misunderstanding and representation noises. To tackle these issues, we propose a novel IMVC based on Debiased and Denoised Representation (DDR-IMVC) learning. Specifically, it utilizes the unbiased representation learned from complete views to refine the biased representation learned from data with missing views. Additionally, we introduce a robust contrastive learning for consensus representation to mitigate cluster collapse risk induced by misalignment noise. Comprehensive experiments demonstrate that DDR-IMVC achieves superior performance compared with state-of-the-art methods.

## 1 Introduction

Multi-view clustering (MVC) has achieved significant breakthroughs in unsupervised learning (Fang et al., 2023; Yan et al., 2024b; Zhang et al., 2024a). Unfortunately, the superior performance of existing MVC methods largely relies on the assumption of complete cross-view samples. However, due to failures during data acquisition or storage, partial data in some views may be unavailable. As a result, existing MVC methods are inapplicable for incomplete multi-view data. Therefore, the problem of incomplete multi-view clustering (IMVC) has attracted increasing attention (Xu et al., 2024; Dong et al., 2024; Wan et al., 2024). The goal of IMVC is to uncover the common clustering patterns hidden in incomplete multi-view data and to group unlabeled instances into distinct clusters.

Existing IMVC methods can be categorized into two types: traditional methods and deep methods. Traditional IMVC methods employ matrix completion to impute missing views for clustering, while some other methods utilize machine learning techniques to learn a shared low-dimensional embedding. These methods can be further divided into three subcategories: kernel-based methods (Liu et al., 2019; 2020), matrix factorization-based methods (Hu & Chen, 2019; Chao et al., 2022), and graph learning-based methods (Wen et al., 2019; Li et al., 2022). However, the performance of traditional IMVC methods largely depend on the quality of features. Although they are more interpretable than deep methods, their representational capacity is often limited.

Owing to the powerful generalization and representation capabilities of deep neural networks, deep IMVC methods have achieved outstanding performance. Some deep methods address the view incomplete problem by imputing the miss-view data with neural network. For example, generative adversarial network (GAN)-based methods employ GAN to produce the missing data (Wang et al., 2021); prototype-matching-based methods (Liu et al., 2022; Jin et al., 2023) learn a certain number of prototypes from the available data and then establish correspondences to recover missing data.

---

*Corresponding Author

However, these methods may incur high computational complexity, and it is generally difficult to achieve accurate missing view recovery, which thus introduces noises to the data and degrade the clustering performance. Differently, some other methods attempt to perform miss-view imputation in the feature space rather than the data space by fusing the latent representation of other views (Wen et al., 2019; Xu et al., 2024). Therefore, these methods effectively reduce the complexity and realize impressive performance. Nevertheless, they overlook inter-view distribution difference, resulting in the distribution shift of the shared representation between the complete views and incomplete ones, which further introduces substantial cross-view misalignment noise and lead to erroneous clustering structures (Lin et al., 2022; Yin et al., 2025).

To address the above issues, we propose a novel incomplete multi-view clustering method based on debiased and denoised representation learning, named DDR-IMVC. The proposed framework is illustrated in Figure 1. Notably, DDR-IMVC is not merely restricted to intra-view or inter-view interactions, but instead enables concurrent interactions across all instances. Specifically, to bridge the semantic gaps across views, DDR-IMVC derives the neighborhood relationships of samples with missing views from their available view information, and leverages these relationships to refine the corresponding missing-view features. In practice, we design adaptive weight matrices based on cluster separability to collaboratively integrate low-dimensional representation from all views and to accommodate the influence of varying degrees of missing instances. In this space, it introduces an attention-based complementarity refinement which utilizes the unbiased representation learned from complete views together with their neighboring relationship to refine the biased representation learned from data with missing views. Additionally, to enhance the consensus representation and eliminate the redundancy of all views, DDR-IMVC maximizes the mutual information between the consensus representation and the view-specific representations. Furthermore, to mitigate misalignment noises in shared representation and improve its discriminativeness, DDR-IMVC employs a denoised contrastive strategy to learn a clear clustering structure and reduce the risk of clustering collapse. Our main contributions can be summarized as follows:

- We propose an innovative incomplete multi-view clustering framework, i.e., DDR-IMVC, which employs unbiased representation to correct and refine the distribution shifts of the biased representation.

- To alleviate the cluster collapse problem induced by misalignment noise, we adopt a robust contrastive constraint based on consensus representations, which facilitates discriminative and robust consensus representation learning.

- We analyze the performance of DDR-IMVC with extensive experiments, whose results demonstrate that DDR-IMVC outperforms state-of-the-art methods across four datasets under varying missing rates.

## 2 RELATED WORK

### 2.1 MULTI-VIEW CLUSTERING

MVC groups samples with similar feature patterns into the same cluster by integrating feature information from different views (Yang & Wang, 2018; Zhou et al., 2024). Deep autoencoders, as powerful feature extraction tools, have been widely applied in MVC. To address the inconsistency between discrete clustering information and continuous visual information, Xu et al. (2021) employs a variational autoencoder to learn disentangled representations. MFLVC relies on an autoencoder to learn latent features at different levels to mine common semantics (Xu et al., 2022b). However, these methods struggle to eliminate the interference of private information and noise during consistent information extraction. Yan et al. (2024a) proposes a novel variational autoencoder under information bottleneck theory to preserve clustering information. Unlike the above approaches that optimize reconstruction loss to learn latent features, Xie et al. (2020) constructs a multi-view joint clustering network using stacked autoencoders, convolutional autoencoders, and variational autoencoders to capture precise multi-view features. Trosten et al. (2021); Tang & Liu (2022) employ encoding networks to extract view-specific features while maintaining cluster compactness through clustering constraints. Moreover, for real-world multi-view data with missing views, the above methods often struggle to uncover accurate data representations. Therefore, uncovering accurate clustering patterns in incomplete multi-view data has become an important research direction.

## 2.2 INCOMPLETE MULTI-VIEW CLUSTERING

In recent years, IMVC has achieved significant breakthroughs. Generally, deep IMVC methods can be divided into four categories: 1) Autoencoder-based methods. Lin et al. (2021; 2022) predict the missing data with the autoencoders to leverage the available data across views, with the goal of minimizing the conditional entropy. 2) Adversarial Network-based methods. Wang et al. (2021) explicitly generates missing view data through generative adversarial networks (GANs) and integrates multi-view information to achieve efficient clustering. Wang et al. (2023) proposes a self-supervised framework that combines GANs with dual contrastive learning, exploiting the hidden information in incomplete data. 3) Graph Convolutional Network-based methods. Tang & Liu (2022) dynamically updates neighbors based on learned semantic features, avoiding the interference of low-quality samples during data completion. Chao et al. (2024) constructs a neighbor-sample adjacency matrix and adopts graph convolutional networks (GCNs) to complete missing samples. Pu et al. (2024) constructs a latent graph to preserve topological information for the dynamic imputation of missing embedded features. Chao et al. (2025) adaptively completes missing representations by integrating intra-view local relationships and cross-view global relationships through GCNs. 4) Prototype-based methods. Dai et al. (2025) proposes an IMVC framework in a common semantic space based on consensus semantics without data completion or alignment. Yuan et al. (2025) introduces a robust prototype contrastive strategy to handle overfitting caused by prototype misalignment. Despite their effectiveness, most IMVC methods ignore the potential inter-view distribution bias due to missing views.

## 3 METHODOLOGY

### 3.1 NOTATIONS

A multi-view dataset $\mathcal{X} = \{\mathbf{X}^v \in \mathbb{R}^{N \times D_v}\}_{v=1}^V$ consists of $N$ samples, each represented by $V$ views of dimensionality $D_v$. There are $N_u$ complete samples with all views and $N_b$ samples with missing views. Let the complete samples be denoted as $\{\mathbf{X}_C^v\}_{v=1}^V$, and the samples with missing views be denoted as $\{\mathbf{X}_I^v\}_{v=1}^V$. A complete view indicator matrix $\mathbf{M} \in \{0,1\}^{N \times V}$ indicates the positions of missing views. $\mathbf{M}_{iv}$ is set to 1 if the $i$-th sample in the $v$-th view is observed; otherwise, it indicates a missing view. It is assumed that no sample is missing in all views simultaneously, i.e., $\forall i \in \{1, \ldots, N\}, \sum_{v=1}^V \mathbf{M}_{iv} \geq 1$. The task is to cluster these $N$ samples with potentially missing views into $K$ clusters.

### 3.2 VIEW-SPECIFIC RECONSTRUCTION

Considering that the data across different views are mostly heterogeneous and differently distributed, we provide independent autoencoders for each view to alleviate clustering instability on the manifold structure in high-dimensional space (Hinton & Salakhutdinov, 2006; Guo et al., 2017). An autoencoder $\mathcal{E}_\theta^v(\cdot)$ is used to learn the embedding of the sample:

$$\mathbf{Z}^v = \mathcal{E}_\theta^v(\mathbf{X}^v), \tag{1}$$

where $\mathbf{Z}^v \in \mathbb{R}^{N \times d}$ denotes the view-specific representation of the $v$-th view in the $d$-dimensional embedding space. $\theta$ represents the learnable parameters of the autoencoder. Then, we reconstruct the embedding $\mathbf{Z}^v$ into $\hat{\mathbf{X}}^v \in \mathbb{R}^{N \times D_v}$ with the decoder $\mathcal{D}_\phi^v(\cdot)$, as follows:

$$\hat{\mathbf{X}}^v = \mathcal{D}_\phi^v(\mathbf{Z}^v), \tag{2}$$

where $\phi$ denotes the learnable parameters of the decoder. The reconstruction loss across all views can be expressed as follows:

$$\mathcal{L}_{\text{REC}} = \sum_{v=1}^V ||\mathbf{X}^v - \hat{\mathbf{X}}^v||_2^2. \tag{3}$$

### 3.3 UNBIASED REFINEMENT FOR BIASED REPRESENTATION

Then, we fuse view-specific representations into a unified common representation via an adaptive weight matrix. In the common embedding space, the common representations contain consistent

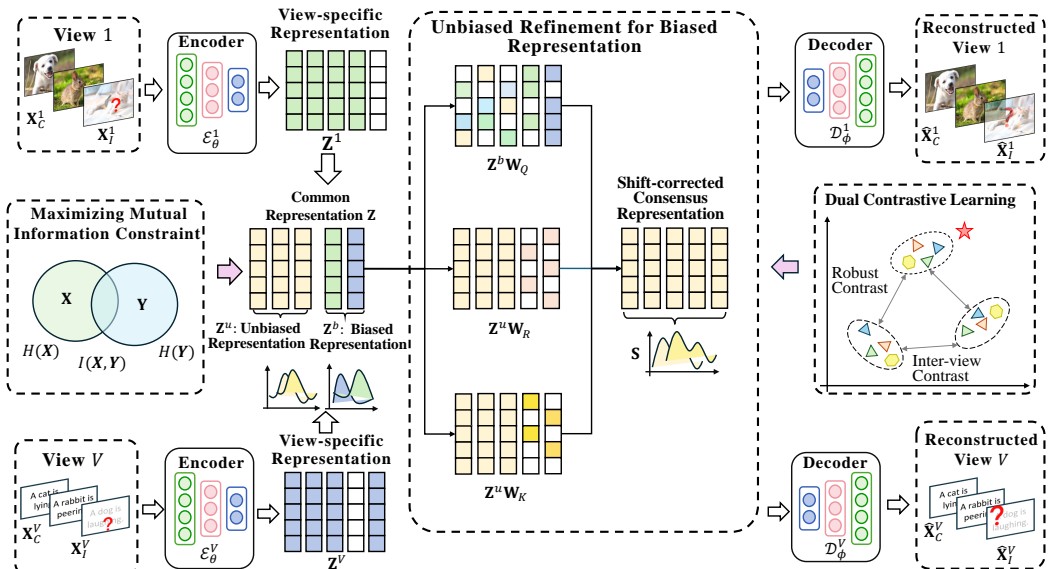

Figure 1: The architecture of our proposed DDR-IMVC framework. (a) Independent autoencoders are employed for each view to extract deep features. (b) The deep features are adaptively projected in a consensus embedding space to bridge the semantic gaps across views. (c) An attention-based refinement strategy is employed to optimize the biased representations introduced by the incomplete sample. (d) A denoised consensus representation contrastive strategy is adopted to alleviate the risk of clustering collapse.

and complementary information. Multi-view complementary information can enhance cluster separability (Zhang et al., 2024b; Dai et al., 2025). Since variance reflects sample deviation from the mean along a certain dimension, with well-separated clusters exhibiting high value (Xu et al., 2023), we leverage the variance to assess the separability of clusters and compute an adaptive weight matrix $\mathbf{W} \in \mathbb{R}^{N \times V}$ that preserves the clustering information of views with well-defined cluster structures:

$$\mathbf{W}_{iv} = \frac{\mathrm{Var}(\mathbf{Z}_C^v)}{\sum_{v'=1}^{V} \mathbf{M}_{iv'} \mathrm{Var}(\mathbf{Z}_C^{v'})}, \tag{4}$$

where $\mathbf{W}_{iv}$ denotes the weight of the $i$-th sample in the $v$-th view. $\mathrm{Var}(\cdot)$ represents the variance operator. The mask $\mathbf{M}_{iv'}$ allows the weight matrix to adapt to missing data. $\mathbf{Z}^v = [\mathbf{Z}_I^v; \mathbf{Z}_C^v]$ denotes representations of the complete samples and incomplete samples in the $v$-th view. Based on the weight matrix $\mathbf{W}$, the samples are mapped to the view-shared common embeddings $\mathbf{Z} \in \mathbb{R}^{N \times d}$.

$$\mathbf{z}_i = \sum_{v=1}^{V} \mathbf{W}_{iv} \mathbf{z}_i^v. \tag{5}$$

where $\mathbf{z}_i \in \mathbf{Z}$ denotes the common representation of the $i$-th sample from all views.

The view-missing issue can cause distribution shifts in the common representation $\mathbf{Z}$. To be specific, the distribution of the common representation fused from the complete views is different from that fused by incomplete views. Thus, we divide the common representation $\mathbf{Z}$ into unbiased representation $\mathbf{Z}^u$ and biased representation $\mathbf{Z}^b$. The unbiased representations are defined as the common representations of samples with complete views, while the biased representations correspond to samples with missing views:

$$\mathbf{Z}^u = \mathbf{Z}\big[\{\, i \mid \prod_{v=1}^{V} \mathbf{M}_{iv} = 1 \,\}, : \big], \tag{6}$$

$$\mathbf{Z}^b = \mathbf{Z}\big[\{\, i \mid \prod_{v=1}^{V} \mathbf{M}_{iv} = 0 \,\}, : \big]. \tag{7}$$

The distribution shift will reduce clustering separability. To address this problem, we innovatively propose an attention-based complementarity refinement. The core idea is to compute the similarity between the biased representations $\mathbf{z}_i^b$ and unbiased representations $\mathbf{z}_i^u$ as sample affinity attention weights $\mathbf{A}$. Then, it extracts the unbiased representations most compatible with the representation with missing views to correct the distribution shifts. Specifically, we first leverage a multi-head attention network to compute the corresponding affinity attention weights as follows:

$$\mathbf{A}^{(l)} = \text{Softmax} \left( \frac{\mathbf{Z}^b \mathbf{W}_Q^{(l)} \left( \mathbf{Z}^u \mathbf{W}_K^{(l)} \right)^\top}{\sqrt{d/L}} \right), l = 1, 2, ..., L \tag{8}$$

where $L$ denotes the number of attention heads, $\mathbf{W}_Q^{(l)}$ and $\mathbf{W}_K^{(l)}$ denote learnable parameters of the $l$-th head, $d$ denotes the dimension of the common representations $\mathbf{Z}$. Based on the affinity attention weights, the unbiased representations can be used to correct the distribution shifts present in the biased representation, and we can obtain the corrected biased representation $\mathbf{B}^{(l)}$ of the $l$-th head as follows:

$$\mathbf{B}^{(l)} = \mathbf{A}^{(l)}(\mathbf{Z}^u \mathbf{W}_R^{(l)}), \tag{9}$$

where $\mathbf{W}_R^{(l)}$ denotes learnable parameters. Finally, the results from all attention heads are concatenated to obtain the corrective biased representation $\mathbf{B}$ in the common embedding space:

$$\mathbf{B} = [\mathbf{B}^{(1)}, ..., \mathbf{B}^{(L)}], \tag{10}$$

Then we incorporate corrective representation $\mathbf{B}$ into the biased representations to obtain shift-corrected consensus representations $\mathbf{S} \in \mathbb{R}^{N \times d}$:

$$\mathbf{S} = [\mathbf{Z}^u; \mathbf{Z}^b] + [\mathbf{0}; \mathbf{B}], \tag{11}$$

where $\mathbf{0}$ denotes an all-zero matrix with the same dimensions as $\mathbf{Z}^u$. Obviously, the effectiveness of correcting the distribution shifts caused by missing views lies in the quality of the common representations, and learning accurate common representations from different views requires exploiting the correlation between the view-specific representations and the common representation. A natural idea is to maximize the mutual information (Lin et al., 2021) between unbiased common representations and view-specific representations to enforce that the unbiased common representations retain as much consensus information. Specifically, we regard the unbiased representations as the anchor of the representations of each view, and sequentially introduce a maximizing mutual information constraint between the anchor and a specific view with the following loss function:

$$\mathcal{L}_{\text{MMI}} = -\sum_{v=1}^{V} \left( I(\mathbf{S}_C; \mathbf{Z}_C^v) + \alpha \left( H(\mathbf{S}_C) + H(\mathbf{Z}_C^v) \right) \right), \tag{12}$$

where $\mathbf{S}_C$ denotes the shift-corrected consensus projections $\mathbf{S}$ corresponding to the sample with complete views, $\alpha$ serves as the entropy regularization coefficient, $I(\cdot)$ denotes mutual information, and $H(\cdot)$ denotes information entropy, which can be computed as follows:

$$H(\mathbf{X}) = -\sum_x p(x) \log p(x), \tag{13}$$

$$I(\mathbf{X}; \mathbf{Y}) = \sum_x \sum_y p(x, y) \log \frac{p(x, y)}{p(x)p(y)}. \tag{14}$$

where $p(x)$ denotes the probability distribution of the random variable $\mathbf{X}$, and $p(x, y)$ denotes the joint distribution of the random variables $\mathbf{X}$ and $\mathbf{Y}$.

Since the consensus representations capture both the inherent information of the incomplete-view samples and the clustering information of its affiliated complete-view samples, restoring embedding with consensus representations can effectively ensure the integrity of the sample structure and the consistency of the distribution. Therefore, we complete the missing-view representations from consensus representations by:

$$\mathbf{S}^v = \mathbf{Z}^v + (1 - \tilde{\mathbf{M}}^v) \odot \mathbf{S}, \tag{15}$$

where $\odot$ denotes the element-wise multiplication. $\mathbf{S}^v$ denotes the representations after completion in the $v$-th view. We take the $v$-th column of matrix $\mathbf{M} \in \mathbb{R}^{N \times V}$ as $\mathbf{m}^v \in \mathbb{R}^{N \times 1}$ that indicates the missing samples in the $v$-th view and expand it into $\tilde{\mathbf{M}}^v \in \mathbb{R}^{N \times d}$ by replicating $\mathbf{m}^v$ $d$ times. Finally, we fuse the completed representations $\mathbf{S}^v$ into $\mathbf{S}'$ via the weight matrix that is constructed by leveraging the variance of the representation from each view.

### 3.4 DUAL CONTRASTIVE LEARNING FOR DENOISED REPRESENTATION

#### 3.4.1 INTER-VIEW CONTRASTIVE LEARNING

To overcome the heterogeneity in incomplete multi-view learning, we construct positive pairs from the same instance across views and negative pairs from different instances. Contrastive learning maximizes the positive-pair correlation while minimizing that of negative pairs. The contrastive learning loss across all views is:

$$\mathcal{L}_{\mathrm{c}} = -\sum_{v=1}^{V} \sum_{\substack{k=1, \\ v \neq k}}^{V} \sum_{i=1}^{N} \log \frac{\exp\left(\mathrm{sim}\left(\mathbf{s}_i^v, \mathbf{s}_i^k\right) / \tau\right)}{\sum_{j=1, j \neq i}^{N} \exp\left(\mathrm{sim}\left(\mathbf{s}_i^v, \mathbf{s}_j^k\right) / \tau\right)}, \tag{16}$$

where $\mathbf{s}_i^v$ and $\mathbf{s}_j^k$ denote the representations from $v$-th view and $k$-th view. $\mathrm{sim}(\cdot)$ is cosine similarity. $\tau$ represents the temperature coefficient.

#### 3.4.2 ROBUST CONTRASTIVE LEARNING

To prevent cluster collapse after completing missing views with consensus representations, it is necessary to impose constraints between the consensus representations and the recovered representations. A conventional approach is to apply contrastive learning between them. Although consensus representations capture the consistency of the data distribution, the completion process inevitably introduces potential noise. Conventional contrastive learning's strong focus on hard samples can exacerbate noise-induced overfitting. Inspired by Yuan et al. (2024), we employ a denoised contrastive learning between consensus representation $\mathbf{S}$ and each view representation $\mathbf{S}^v$ to enhance the robustness of consensus representations against noise.

Set $f(\mathbf{s}_i, \mathbf{s}_j) = \frac{\exp(\mathrm{sim}(\mathbf{s}_i, \mathbf{s}_j)/\tau)}{\sum_{n=1}^{N} \exp(\mathrm{sim}(\mathbf{s}_i, \mathbf{s}_n)/\tau)}$, and thus $\sum_{j=1}^{N} f(\mathbf{s}_i, \mathbf{s}_j) = 1$. For the general form of InfoNCE, its power series expansion over the interval $[0, 1]$ is:

$$\begin{aligned}
\mathcal{L}_{\mathrm{Info}} &= -\frac{1}{N} \sum_{v=1}^{V} \sum_{i=1}^{N} \log \frac{\exp\left(\mathrm{sim}(\mathbf{s}_i, \mathbf{s}_i^v)/\tau\right)}{\sum_{n=1}^{N} \exp\left(\mathrm{sim}(\mathbf{s}_i, \mathbf{s}_n^v)/\tau\right)} \\
&= \frac{1}{N} \sum_{v=1}^{V} \sum_{i=1}^{N} \left[ (1 - f(\mathbf{s}_i, \mathbf{s}_i^v)) + \frac{(1 - f(\mathbf{s}_i, \mathbf{s}_i^v))^2}{2} + ... + \frac{(1 - f(\mathbf{s}_i, \mathbf{s}_i^v))^c}{c} + ... \right] \\
&= \frac{1}{N} \sum_{v=1}^{V} \sum_{i=1}^{N} \left[ \frac{1}{2}(2 - 2f(\mathbf{s}_i, \mathbf{s}_i^v)) + \frac{(1 - f(\mathbf{s}_i, \mathbf{s}_i^v))^2}{2} + ... + \frac{(1 - f(\mathbf{s}_i, \mathbf{s}_i^v))^c}{c} + ... \right] \\
&= \frac{1}{N} \sum_{v=1}^{V} \sum_{i=1}^{N} \left[ \frac{1}{2}\|\mathbf{e}_i - \mathbf{f}_i\|_1 + \frac{(1 - f(\mathbf{s}_i, \mathbf{s}_i^v))^2}{2} + ... + \frac{(1 - f(\mathbf{s}_i, \mathbf{s}_i^v))^c}{c} + ... \right] \\
&= \frac{1}{2N} \sum_{v=1}^{V} \sum_{i=1}^{N} \|\mathbf{e}_i - \mathbf{f}_i\|_1 + \frac{1}{N} \sum_{v=1}^{V} \sum_{i=1}^{N} \sum_{c=2}^{\infty} \frac{(1 - f(\mathbf{s}_i, \mathbf{s}_i^v))^c}{c}
\end{aligned} \tag{17}$$

where $\mathbf{e}_i$ denotes the one-hot encoding whose $i$-th element is 1; $\mathbf{f}_i$ is a vector whose $j$-th element is $f(\mathbf{s}_i, \mathbf{s}_i^v)$. It can be seen that, after expanding InfoNCE into an infinite series, the first term is exactly the Mean Absolute Error (MAE) loss, which is proven to be robust to noise (Ghosh et al., 2015; 2017). However, MAE loss treats each sample equally. The infinite terms can provide differentiated attention to samples but are sensitive to noise. Therefore, we can construct a robust contrastive loss by truncating part of the infinite series to maintain a balance between MAE loss and InfoNCE loss, which is adjusted by a truncation coefficient $C$. Specifically, we take the first $C$ terms of the infinite series and obtain the robust contrastive loss as follows:

$$\mathcal{L}_{\mathrm{r}} = \frac{1}{N} \sum_{v=1}^{V} \sum_{i=1}^{N} \sum_{c=1}^{C} \frac{(1 - f(\mathbf{s}_i, \mathbf{s}_i^v))^c}{c}, \tag{18}$$

Its significance lies in that it transforms the unbounded amplification of $-\log f(\mathbf{s}_i, \mathbf{s}_i^v)$ for hard samples into a bounded approximation, balancing positive sample discrimination and noise suppression. Adjusting the truncation coefficient $C$ allows tuning between cluster collapse and noise robustness.

---

**Algorithm 1** DDP for Incomplete Multi-view Clustering

---

1: **Input**: Incomplete multi-view dataset $\mathcal{X} = \{\mathbf{X}^v\}_{v=1}^V$ for all $N$ samples, Training epoch $E$, Hyper-parameter $\lambda_1$, $\lambda_2$, $\alpha$, and $C$.
2: Construct the complete view indicator matrix $\mathbf{M} \in \mathbb{R}^{N \times V}$.
3: **while** Not reaching epochs $E$ **do**
4:     Calculate the embedding representation $\{\mathbf{Z}^v\}_{v=1}^V$ by Eq.(1).
5:     Calculate the common representations $\mathbf{Z}$ by Eq.(5).
6:     Correct the shifts to obtain consensus representations $\mathbf{S}$ by Eq.(11).
7:     Impute the each view embeddings $\mathbf{S}^v$ by Eq.(15).
8:     Compute the clustering-friendly representation $\mathbf{S}'$ with the imputed embeddings.
9:     Optimize the total loss function $\mathcal{L}_{all}$ by Eq.(20).
10: **end while**
11: Perform k-means clustering algorithm on $\mathbf{S}'$.
12: **Output**: K clusters for $N$ samples.

---

Finally, the dual contrastive loss is:

$$\mathcal{L}_{\text{DCL}} = \mathcal{L}_{\text{c}} + \mathcal{L}_{\text{r}}. \tag{19}$$

## 3.5 The Objective Function

Overall, the total loss function of our method consists of three parts is formulated as:

$$\mathcal{L}_{all} = \mathcal{L}_{\text{REC}} + \lambda_1 \mathcal{L}_{\text{MMI}} + \lambda_2 \mathcal{L}_{\text{DCL}}. \tag{20}$$

$\lambda_1$ and $\lambda_2$ are trade-off parameters. $\mathcal{L}_{\text{REC}}$ is the autoencoder reconstruction loss. $\mathcal{L}_{\text{MMI}}$ is the maximum mutual information loss, used to enhance the common cluster information. $\mathcal{L}_{\text{DCL}}$ is the dual contrastive loss that mitigates heterogeneity in incomplete multi-view learning and prevents cluster collapse. Finally, K-means is performed on $\mathbf{S}' \in \mathbb{R}^{N \times d}$ to obtain $K$ clusters.

## 4 Experiments

### 4.1 Datasets

We conducted experiments on four representative datasets. The datasets are: **HandWritten** (LeCun et al., 1989) comprises 2,100 samples belonging to 10 categories corresponding to digits from 0 to 9. We employ three distinct features Pixel, Fourier and Profile for analysis. **Scene-15** (Fei-Fei & Perona, 2005) consists of 15 categories with a total of 4,485 samples. GIST, PHOG, and LBP are selected as three views in our experiments. **ALOI-100** (Geusebroek et al., 2005) contains 10,800 object images belonging to 100 categories. We extract HSB, RGB, Colorsim, and Haralick features to construct multi-view data. **LandUse-21** (Yang & Newsam, 2010) comprises 2,100 samples belonging to 21 categories corresponding to different land-use scene categories. GIST, PHOG, and LBP are used for analysis. To evaluate the performance of our approach, we employ three standard metrics: Accuracy (ACC), Normalized Mutual Information (NMI), and Adjusted Rand Index (ARI).

### 4.2 Compare Method

DDR-IMVC is compared with nine SOTA methods. **Fusion-kmeans** clusters the mean-fused features with k-means. **Completer** (Lin et al., 2021) predicts missing views by minimizing conditional entropy. **DIMVC** (Xu et al., 2022a) proposes a no-imputation framework that maps data to reveal linear separability. **DSIMVC** (Tang & Liu, 2022) completes views by dynamically mining semantic features of neighbors. **DCP** (Lin et al., 2022) learns consistent representations via dual contrastive learning under the information-theoretic framework. **ProImp** (Li et al., 2023) recovers data by learning prototypes with dual attention layers. **APADC** (Xu et al., 2023) achieves imputation-free strategy through adaptive projection and distribution alignment. **ICMVC**(Chao et al., 2024) completes missing views with GCNs and aligns distributions through high confidence guidance. **GHICMC** (Chao et al., 2025) employs cascaded GCNs to enable global graph propagation and hierarchical information transfer.

Table 1: Clustering results of all methods on four datasets. The best and second-best results are highlighted with bold and underline, respectively.

| | Missing_rates | 0.1 | | | 0.3 | | | 0.5 | | | 0.7 | | |
|---|---|---|---|---|---|---|---|---|---|---|---|---|---|
| | Metrics | ACC | NMI | ARI | ACC | NMI | ARI | ACC | NMI | ARI | ACC | NMI | ARI |
| LandUse-21 | Fusion-kmeans | 20.45 | 25.42 | 8.62 | 16.41 | 17.48 | 5.51 | 12.86 | 12.58 | 2.77 | 11.45 | 9.81 | 1.38 |
| | Completer(2021) | 26.40 | 32.48 | 13.93 | 26.96 | 32.64 | 12.09 | 21.36 | 26.27 | 9.34 | 24.43 | 29.01 | 10.31 |
| | DIMVC(2022) | 24.63 | 30.04 | 10.58 | 23.69 | 29.94 | 10.01 | 22.40 | 27.78 | 9.38 | 21.77 | 26.14 | 7.91 |
| | DSIMVC(2022) | 18.47 | 19.34 | 5.58 | 17.95 | 18.47 | 5.16 | 18.13 | 18.53 | 5.26 | 17.90 | 17.97 | 5.11 |
| | DCP(2023) | 26.78 | 30.87 | 13.80 | 27.08 | 30.69 | 13.80 | 23.07 | 27.00 | 11.31 | 25.18 | 28.04 | 12.00 |
| | ProImp(2023) | 22.38 | 23.79 | 8.76 | 19.53 | 20.55 | 6.86 | 20.30 | 21.94 | 7.32 | 15.10 | 15.48 | 4.00 |
| | APADC(2023) | 22.75 | 31.90 | 9.50 | 18.08 | 24.72 | 7.22 | 15.67 | 21.23 | 5.61 | 15.11 | 20.08 | 4.84 |
| | ICMVC(2024) | 28.18 | 31.78 | 15.14 | 25.77 | 29.39 | 12.69 | 25.98 | 27.74 | 11.92 | 22.26 | 24.95 | 9.31 |
| | GHICMC(2025) | 26.86 | 31.14 | 12.81 | 25.72 | 29.26 | 11.32 | 25.15 | 28.57 | 11.26 | 23.53 | 26.55 | 9.72 |
| | **Ours** | **28.21** | **34.68** | 14.93 | **28.02** | **33.49** | **14.27** | **27.14** | **31.89** | **13.24** | **25.36** | 28.09 | 10.67 |
| Scene-15 | Fusion-kmeans | 34.20 | 34.84 | 20.93 | 22.39 | 23.60 | 12.80 | 17.54 | 17.31 | 7.88 | 14.90 | 13.54 | 4.29 |
| | Completer(2021) | 40.28 | 42.50 | 23.13 | 40.12 | 42.93 | 23.96 | 39.12 | 41.79 | 22.98 | 38.05 | 40.22 | 21.84 |
| | DIMVC(2022) | 32.95 | 27.41 | 15.61 | 33.51 | 29.42 | 16.75 | 30.65 | 25.21 | 13.64 | 29.58 | 24.11 | 12.85 |
| | DSIMVC(2022) | 27.65 | 29.74 | 14.11 | 26.73 | 29.36 | 13.94 | 26.40 | 28.03 | 13.04 | 25.31 | 27.04 | 12.43 |
| | DCP(2023) | 38.54 | 42.39 | 23.33 | 40.49 | 43.10 | 24.14 | 39.50 | 42.35 | 23.51 | 38.55 | 40.57 | 21.72 |
| | ProImp(2023) | 40.74 | 42.14 | 24.00 | 41.69 | 43.03 | 25.28 | 40.28 | 41.80 | 23.89 | 39.96 | 40.35 | 22.92 |
| | APADC(2023) | 43.70 | 44.20 | 26.00 | 41.80 | 43.10 | 24.30 | 39.90 | 42.40 | 23.80 | 38.50 | 41.10 | 22.80 |
| | ICMVC(2024) | 38.78 | 36.62 | 21.84 | 37.40 | 34.94 | 20.60 | 31.35 | 27.91 | 14.98 | 25.31 | 23.88 | 11.33 |
| | GHICMC(2025) | 41.26 | 43.21 | 24.95 | 40.98 | 43.13 | 25.05 | 40.91 | 42.29 | 24.64 | 38.79 | 40.94 | 22.92 |
| | **Ours** | **46.16** | **47.62** | **28.69** | **45.53** | **45.99** | **28.05** | **44.35** | **43.67** | **26.79** | **42.12** | **41.20** | **24.73** |
| HandWritten | Fusion-kmeans | 41.70 | 47.59 | 34.22 | 36.28 | 38.76 | 21.46 | 29.64 | 28.51 | 11.54 | 25.69 | 22.14 | 6.58 |
| | Completer(2021) | 83.22 | 82.47 | 73.59 | 75.38 | 77.55 | 61.69 | 74.05 | 76.13 | 58.89 | 78.55 | 76.07 | 68.67 |
| | DIMVC(2022) | 67.13 | 63.17 | 53.16 | 59.43 | 56.49 | 43.19 | 54.80 | 50.50 | 30.76 | 43.82 | 41.54 | 23.80 |
| | DSIMVC(2022) | 84.35 | 80.32 | 74.38 | 85.64 | 80.71 | 75.94 | 84.73 | 78.82 | 74.13 | 82.71 | 75.35 | 69.85 |
| | DCP(2023) | 53.35 | 65.72 | 35.60 | 51.96 | 63.88 | 31.49 | 59.06 | 65.07 | 36.51 | 60.97 | 60.53 | 29.90 |
| | ProImp(2023) | 83.20 | 80.29 | 74.17 | 84.24 | 77.75 | 72.60 | 78.16 | 70.79 | 63.96 | 80.31 | 68.85 | 62.92 |
| | APADC(2023) | 67.43 | 65.34 | 47.18 | 68.95 | 67.28 | 45.98 | 68.85 | 68.61 | 56.43 | 61.77 | 61.97 | 48.26 |
| | ICMVC(2024) | 83.16 | 81.33 | 74.78 | 82.01 | 79.62 | 72.22 | 75.13 | 71.99 | 63.19 | 72.47 | 70.01 | 59.71 |
| | GHICMC(2025) | 96.19 | 92.14 | 92.89 | 96.11 | 91.32 | 90.83 | **94.88** | **89.16** | **89.10** | **92.73** | **85.85** | **84.71** |
| | **Ours** | **96.38** | **92.23** | 91.99 | **96.15** | **91.49** | **91.21** | 94.34 | 88.38 | 87.87 | 90.86 | 82.65 | 81.92 |
| ALOI-100 | Fusion-kmeans | 52.37 | 72.31 | 40.79 | 30.63 | 55.28 | 16.10 | 22.48 | 47.33 | 7.46 | 17.39 | 41.82 | 4.94 |
| | Completer(2021) | 48.19 | 77.96 | 44.25 | 43.03 | 72.43 | 36.73 | 36.16 | 66.89 | 26.52 | 34.55 | 64.06 | 24.97 |
| | DIMVC(2022) | 71.86 | 84.99 | 61.79 | 68.52 | 82.15 | 58.31 | 64.80 | 78.53 | 51.36 | 61.64 | 75.33 | 47.25 |
| | DSIMVC(2022) | 38.76 | 67.49 | 29.71 | 38.89 | 66.00 | 29.12 | 39.32 | 64.42 | 28.53 | 35.98 | 61.28 | 25.16 |
| | DCP(2023) | 51.85 | 74.88 | 42.73 | 47.38 | 70.54 | 38.38 | 42.37 | 66.30 | 32.36 | 36.02 | 60.75 | 25.40 |
| | ProImp(2023) | 68.39 | 83.47 | 62.08 | 45.98 | 73.01 | 38.53 | 32.71 | 65.74 | 24.76 | 29.23 | 62.08 | 19.46 |
| | APADC(2023) | 47.40 | 68.92 | 35.02 | 38.95 | 62.27 | 26.10 | 32.78 | 58.16 | 20.10 | 26.02 | 53.91 | 14.11 |
| | ICMVC(2024) | 68.02 | 80.78 | 56.64 | 68.14 | 80.40 | 55.94 | 67.68 | 78.92 | 53.92 | 49.15 | 70.50 | 38.45 |
| | GHICMC(2025) | | OOM | | | OOM | | | OOM | | | OOM | |
| | **Ours** | **76.02** | **88.35** | **67.82** | **73.18** | **85.82** | **64.36** | **69.87** | **82.34** | **58.09** | **66.94** | **78.60** | **53.20** |

Table 2: Ablation study results on LandUse21 and Scene-15 datasets with missing rate 0.3.

| Datasets | | | LandUse21 | | | Scene-15 | | |
|---|---|---|---|---|---|---|---|---|
| $\mathcal{L}_{\text{REC}}$ | $\mathcal{L}_{MMI}$ | $\mathcal{F}_{DCL}$ | ACC | NMI | ARI | ACC | NMI | ARI |
| ✓ | | ✓ | 17.54 | 22.97 | 6.08 | 36.06 | 43.69 | 21.88 |
| ✓ | ✓ | | 24.16 | 26.09 | 11.06 | 41.96 | 40.00 | 25.94 |
| ✓ | | | 16.78 | 17.96 | 5.63 | 21.53 | 21.61 | 11.48 |
| ✓ | ✓ | ✓ | **28.02** | **33.49** | **14.27** | **45.53** | **45.99** | **28.05** |

## 4.3 EXPERIMENTAL SETTINGS

We adopt Adam to optimize our framework DDR-IMVC and all the experiments are conducted in PyTorch 1.13.1 on Windows with an NVIDIA 4070 SUPER GPU. The dimensions of encoders are $D_v$–1024–1024–1024–128. The decoder is symmetric to its corresponding encoder. The number of heads $L$ in multi-head attention is set to 4. The entropy regularization coefficient $\alpha$ is set to 10, and the truncation coefficient $C$ is set to 9.

## 4.4 INCOMPLETE MULTI-VIEW CLUSTERING PERFORMANCE

Table 1 reports the incomplete multi-view clustering results of all methods under different missing rates. It shows that DDR-IMVC can effectively handle high missing rates and large-scale issues in

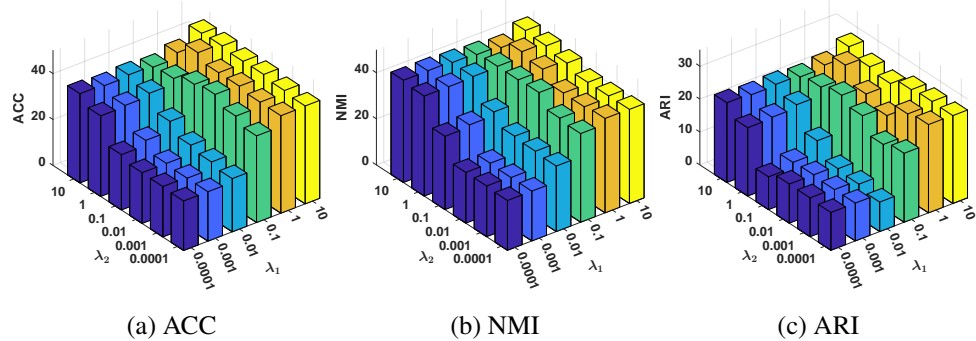

Figure 2: Parameter sensitivity analysis on Scene-15 with the missing rate 0.3.

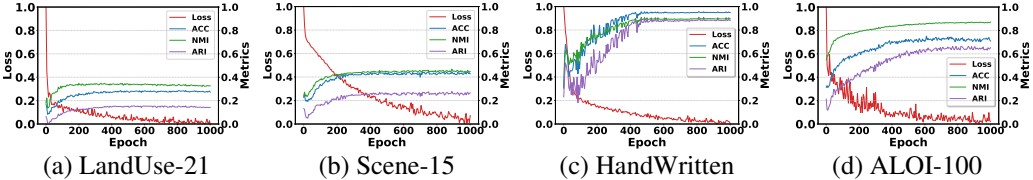

Figure 3: The convergence analysis on all datasets with the missing rate 0.3.

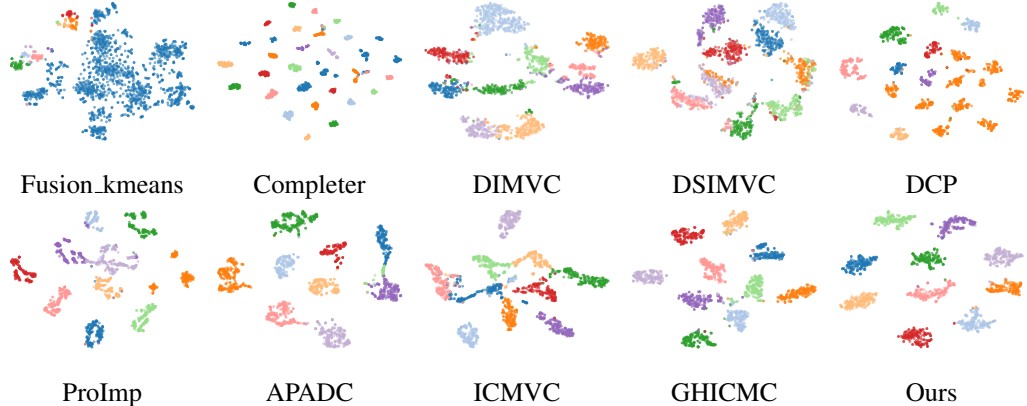

Figure 4: The visualization results of HandWritten dataset in all methods with a missing rate of 0.5.

IMVC. From the perspective of effectiveness, our method significantly outperforms SOTA methods across four datasets. For example, in the Scene-15 dataset, ACC, NMI, and ARI outperform the second-best method, APADC, by an average of 3.56%, 1.92%, and 2.86%, respectively. We notice that when the missing rate is 0.5 or 0.7, DDR-IMVC performs slightly worse than GHICMC. We attribute this to the simplicity of the HandWritten dataset, where inter-class features are singular. Under high missing rates, it is suitable to use cascade graphs for data recovery. For other complex datasets, GHICMC shows a significant performance drop. More critically, its high memory consumption prevents it from handling large-scale datasets. From the perspective of robustness, DDR-IMVC can still maintain a high level of performance under a high missing rate. Moreover, unlike some methods whose performance drops sharply, DDR-IMVC remains stable even as the missing rate increases.

### 4.5 MODEL DISCUSSION

*1) Ablation Study:* To investigate the importance of each component, we conducted an ablation study on our DDR-IMVC framework using LandUse21 and Scene-15 datasets under a missing rate of 0.3. As shown in Table 2, removing the attention correction mechanism for the learned consensus

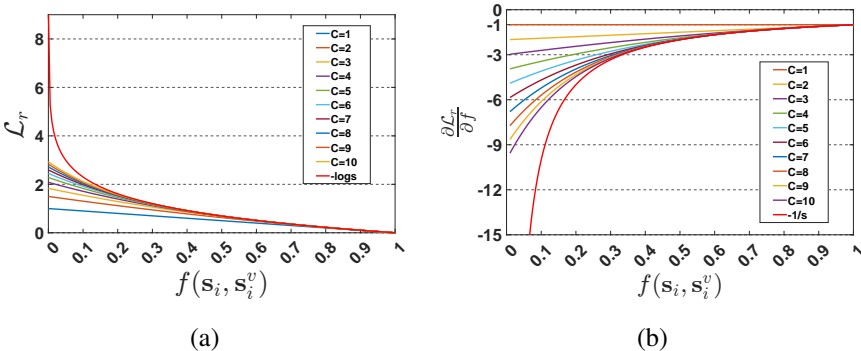

Figure 5: The variation trend of the loss and its gradient.

representations ($\mathcal{L}_{\text{MMI}}$) or the robust contrastive learning ($\mathcal{L}_{\text{DCL}}$) leads to suboptimal performance. When all strategies are applied together, we achieve the best results. The experimental findings demonstrate that the attention correction mechanism can refine the distribution of biased representations, while the robust contrastive learning enhances the consistency of consensus representations and alleviates the risk of cluster collapse.

*2) Parameter Analysis:* Our objective loss mainly involves two trade-off parameters, $\lambda_1$ and $\lambda_2$. To verify their effectiveness, we conducted a parameter analysis by setting both parameters in the range from $10^{-4}$ to 10. As shown in Figure 2, excessively high or low parameter values are unfavorable for clustering. Based on our parameter experiments, we recommend setting the parameter range between 1 and 10.

*3) Convergence Analysis:* Meanwhile, to better verify the convergence and robustness of our model, we observed the convergence performance of all datasets under a 0.3 missing rate. As shown in the Figure 3, the total loss function involved in our training achieved excellent convergence. With the increase in training epochs, various metrics for evaluating clustering also tended to converge.

*4) Visualization Analysis:* As shown in Figure 4, with a missing rate of 0.5, we visualized the distribution of common embedding of all methods on the HandWritten dataset using t-SNE. Through the correction of the distribution shift of missing samples during the imputation process by DDR-IMVC, our method is facilitated to discover the common clustering patterns of all views in the common embedding space.

*5) Discussion of Robust Contrastive Loss:* In Figure 5, we plot the InfoNCE loss and the loss function in Equation (18), as well as their gradients. As described in Section 4.5, the single-sample InfoNCE function is $\mathcal{L}_{info} = -\log f$, with gradient $\frac{\partial \mathcal{L}_{info}}{\partial f} = -\frac{1}{f}$. The function in Equation (18) is $\mathcal{L}_r = \sum_{c=1}^{C} \frac{(1-f)^c}{c}$, with gradient $\frac{\partial \mathcal{L}_r}{\partial f} = -\sum_{c=1}^{C} (1-f)^{c-1}$. When $C = 1$, the gradient of Equation (18) is $-1$, indicating that it treats all samples equally, equivalent to MAE. When $C \to \infty$, it degenerates to InfoNCE, giving excessively high attention to noisy samples. Our loss gradient is smaller than MAE, which indicates that our loss can assign different attention levels to different samples, improving training efficiency. It is larger than InfoNCE and has an upper bound, indicating that our loss prioritizes clean samples, thereby mitigating the issue of excessive attention to noisy samples and enhancing robustness.

## 5    CONCLUSION

In this work, we propose a consensus representation refinement strategy for IMVC to address data shift and misalignment noise introduced by missing views. An adaptive representation fusion constructs a common space. Within this space, unbiased representations correct the distribution of biased representations through an attention mechanism to form robust consensus representations. In addition, we employ a denoising contrastive strategy to prevent cluster collapse that may occur when completing missing views in the consensus representations. The effective combination of these two strategies enables DDR-IMVC to achieve superior performance across several IMVC tasks.

## 6 ACKNOWLEDGMENTS

This work is supported by the National Natural Science Foundation of China under Grant 62176203, the Fundamental Research Funds for the Central Universities (ZYTS25267, QTZX25004), and the Science and Technology Project of Xi'an (Grant 2022JH-JSYF-0009), Open Project of Anhui Provincial Key Laboratory of Multimodal Cognitive Computation, Anhui University (No. MMC202416), Selected Support Project for Scientific and Technological Activities of Returned Overseas Chinese Scholars in Shaanxi Province 2023-02.

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
