# OpenReview forum: "Debiased and Denoised Representation Learning for Incomplete Multi-view Clustering"
_ICLR.cc/2026/Conference — ICLR 2026 Poster_

### Official Review · Reviewer_knnk · 2025-10-31

**Soundness:** 3
**Presentation:** 3
**Contribution:** 3
**Rating:** 6
**Confidence:** 4

**Summary:**

This paper introduces PDD (Projection Debiasing and Denoising), a framework for Incomplete Multi-View Clustering (IMVC). The framework addresses challenges such as distribution shifts caused by missing views and cluster collapse due to misalignment noise. By employing unbiased projection refinement and denoised contrastive learning, PDD achieves SOTA performance on multiple datasets under varying missing rates.

**Strengths:**

1.The use of unbiased projection refinement corrects distribution shifts in biased projections, enhancing consensus embeddings.
2. The framework is robust under varying missing-view rates (10%–70%), outperforming nine strong baselines.

**Weaknesses:**

(1) It is not straightforward to learn the denoising effect of denoising contrastive loss, which should be explained.
(2) It is not clear why the model imputes the missing views in both the representation space and sample space.

**Questions:**

(1) Is the proposed model suitable for large-scale datasets?
(2) Does the proposed model perform stably across all missing rates?

---

> ### Author Response · Authors · 2025-11-24
>
> Thank you for your comments.
>
> **Q1.	It is not straightforward to learn the denoising effect of denoising contrastive loss, which should be explained.**
>
> **R**: Although the consensus projection reflects the consistency of the data distribution, after imputing missing samples, it inevitably introduces noise compared with the original data. Traditional contrastive loss emphasizes noisy samples and can exacerbate the risk of overfitting due to noise. To address this issue, we introduce a denoising contrastive loss that can be considered as the truncated Taylor series expansion of InfoNCE loss adjusted by a truncation coefficient $C$. The first term of the series is Mean Absolute Error (MAE) loss [1], which is proven to be robust to noise [2]. By controlling the truncation coefficient $C$, we could adjust the robustness. Therefore, it is denoising contrastive loss.
>
> [1] Zhang, Zhilu, and Mert Sabuncu. "Generalized cross entropy loss for training deep neural networks with noisy labels." Advances in neural information processing systems 31, 2018.
> [2] Aritra Ghosh, Himanshu Kumar, and PS Sastry. Robust loss functions under label noise for
> deep neural networks. In AAAI, pages 1919–1925, 2017.
>
>
> **Q2. It is not clear why the model imputes the missing views in both the representation space and sample space.**
>
> **R**: We do not perform missing view completion in the sample space. After obtaining the high-quality consensus projection $Z_{E}$ based on the attention mechanism (Eq. 10), we only address the IMVC problem through projection optimization and consensus learning in the feature space.
>
> **Q3. Is the proposed model suitable for large-scale datasets?**
>
> **R**: Unlike some feature-completion methods, our model does not require loading the entire dataset into memory at once. Instead, it processes data in batches, which enables it to scale effectively to large-scale datasets. During the experimental phase, we also take the performance on large datasets into consideration, i.e., ALOI-100. This dataset contains 10,800 samples and as many as 100 categories. Across four missing-view rates on this dataset, our model still achieved superior performance.
>
> **Q4. Does the proposed model perform stably across all missing rates?**
>
> **R**: We evaluate the performance of our method on the Scene-15 dataset with missing rage ranging from 0.1 to 0.7, as illustrated below. It can be observed that our model exhibits stability under a wide range of missing rates.
> | ACC/Missrate |  0.1  |  0.2  |  0.3  |  0.4  |  0.5  |  0.6  |  0.7  |
> | ---------- | --- | --- | --- | --- | ---| --- | --- |
> |   Scene-15   | 46.16 | 45.50 | 45.53 | 45.00 | 44.35 | 43.84 | 42.12 |

---

### Official Review · Reviewer_qR7X · 2025-10-31

**Soundness:** 3
**Presentation:** 3
**Contribution:** 3
**Rating:** 6
**Confidence:** 5

**Summary:**

The paper proposes a novel incomplete multi-view clustering framework. An adaptive projection matrix is constructed to integrate complementary information across views. Then, through an attention-based strategy, the framework utilizes unbiased projection to correct the distribution shift of biased projections, thereby obtaining a bias-corrected consensus projection. By maximizing the mutual information, it enhances cross-view semantic correlations. Finally, an innovative denoising contrastive learning strategy is introduced to alleviate the risk of clustering collapse caused by multi-view heterogeneity and misaligned noise.

**Strengths:**

1.This paper proposes a novel framework named PDD based on debiasing and denoising projection learning. By combining attention mechanisms with contrastive learning, it effectively addresses distribution shifts and clustering collapse caused by missing views in incomplete multi-view clustering.

2.It introduces a denoising contrastive loss based on truncated power series approximation, which transforms the unbounded amplification of hard samples in the InfoNCE loss into a bounded constraint, balancing positive sample discriminability and noise suppression.

**Weaknesses:**

1. There lacks a convergence analysis.
2. In addition to \lambda_{1} and \lambda_{2}, there are also hyperparameters $\alpha$ and $C$, for which the authors do not provide detailed explanations regarding the principles behind their selection.
3. This paper proposes a projection matrix based on prior separability, using variance to measure the projection weights of each view. More expanations should be provided on whether this could lead to imbalanced projections across views.

**Questions:**

1.Can the model converge stably? It is recommended that the authors add convergence analysis experiments to demonstrate the stability and convergence of the proposed algorithm.

2.What are the selection principles for the hyperparameters $\alpha$ and $C$ in this paper? Please provide a detailed explanation.

3.Please provide an explanation for the issue mentioned in weakness 3.

---

> ### Author Response · Authors · 2025-11-24
> **Response to Reviewer qR7X**
>
> Thank you for your comments.
>
> **Q1.	Can the model converge stably? It is recommended that the authors add convergence analysis experiments to demonstrate the stability and convergence of the proposed algorithm.**
>
> **R**: We have conducted the convergence experiments and add the results in the revised manuscript as shown in Figure 3. According the convergence curve, our method is able to converge stably.
>
> **Q2.	What are the selection principles for the hyperparameters and in this paper? Please provide a detailed explanation.**
>
> **R**: The main hyperparameters used in our work are $\lambda_{1}$ and $\lambda_{2}$, and we utilize the grid search strategy and establish the optimal values of them from the interval $[0.0001, 100]$.
>
> **Q3.	More explanations should be provided on whether using variance to measure the projection weights of each view could lead to imbalanced projections across views.**
>
> **R**: Employing variance to measure projection weights aims to balance the contribution of each view based on their separability and will not lead to imbalanced projections. This is because variance can reflect the data separability, and a higher variance means a view tends to have better cluster structure. Therefore, we compute weights from variances, which assigns higher weights for views with clearer cluster structure. Compared with common methods such as mean-based fusion, our approach allows the fusion coefficients to adapt to the data, effectively balancing the projections between views with well-structured clusters.

---

### Official Review · Reviewer_HWFR · 2025-11-01

**Soundness:** 3
**Presentation:** 3
**Contribution:** 4
**Rating:** 6
**Confidence:** 5

**Summary:**

This paper addresses distribution shift in incomplete multi-view clustering caused by missing data. To mitigate cross-view semantic gaps, it optimizes projections into a shared embedding space from an information-theoretic perspective. The method leverages an unbiased projection together with an attention-based refinement strategy to build a robust consensus embedding, thereby alleviating heterogeneity and inconsistency issues commonly encountered in incomplete multi-view clustering.

**Strengths:**

(1) Dealing with data distribution drift is interesting and valuable in incomplete multi-view clustering caused by missing data.
(2) The paper proposes the novel idea of utilizing representations from complete views to refine those from missing views, guided by correlations between complete-view samples and missing-view samples.
(3) The manuscript is clearly written and well organized, providing strong motivation, detailed methodology, and thorough analysis.
(4) Experimental results across multiple datasets demonstrate strong performance.

**Weaknesses:**

1.	Section 2.2 lacks sufficient coverage of recent incomplete multi-view clustering works, particularly those based on missing-view imputation.
2.	The framework illustration (Figure 1) is relatively simple and does not fully reflect key methodological details.
3.	One core contribution is using unbiased projections to correct biased ones; however, a more detailed explanation of its motivation and effectiveness is still expected.

**Questions:**

1. Dealing with data distribution drift is interesting in incomplete multi-view clustering, but how to prove the proposed method to handle the distribution drift better than the previous competitors?
2. Abstract claim that "Consensus representation imputation methods ignore the interview distribution bias due to missing views", what is the interview distribution bias? This concept may require more detailed visualizations to help understanding.

---

> ### Author Response · Authors · 2025-11-24
> **Response to Reviewer HWFR**
>
> Thank you for your comments.
>
> **Q1.	Section 2.2 lacks sufficient coverage of recent incomplete multi-view clustering works, particularly those based on missing-view imputation.**
>
> **R**: In the Related Work section, some works are concerning missing view imputation, such as Completer (Lin et al. 2021), PMIMC (Yuan et al. 2025), DCP (Lin et al. 2022), and CIMIC-GAN (Wang et al. 2023). Following the comment, we have investigated more latest missing view imputation methods (Ref 1,2) and analyzed them in the revised version.
>
> Ref:
> [1] Chao G, Xu K, Xie X, et al. Global Graph Propagation with Hierarchical Information Transfer for Incomplete Contrastive Multi-view Clustering[C]//Proceedings of the AAAI Conference on Artificial Intelligence. 2025, 39(15): 15713-15721.
> [2] Pu J, Cui C, Chen X, et al. Adaptive feature imputation with latent graph for deep incomplete multi-view clustering[C]//Proceedings of the AAAI conference on artificial intelligence. 2024, 38(13): 14633-14641.
>
> **Q2.	The framework illustration (Figure 1) is relatively simple and does not fully reflect key methodological details.**
>
> **R**: We have optimized the framework diagram in the revised manuscript by providing more details.
>
> **Q3.	One core contribution is using unbiased projections to correct biased ones; however, a more detailed explanation of its motivation and effectiveness is still expected.**
>
> **R**: Most existing incomplete multi-view clustering methods utilize the obtained views to impute the projection of missing views, so as to learn the shared projection. We named these projections as biased projection, because their distribution deviates from the projection distribution learned from the complete views. To correct the biased projection, we compute an affinity-based attention weight between the unbiased projections and the biased projections. Then, we select the most relevant unbiased projections and fuse them with the biased projection based on the computed attention weight. Thus, the distribution of the corrective projections is similar to that of the unbiased ones.
>
> **Q4.	How to prove the proposed method to handle the distribution drift better than the previous competitors?**
>
> **R**: We compare our method with methods that employ existing views to impute missing views by visualizing the distribution of learned projections. Specifically, we visualized those distribution on Handwritten dataset at a missing rate of 0.5 and the visualized figures have been added to Figure 5 in the revised manuscript. From the figures, we could see that our method achieves a clearer cluster structure by correcting the distribution shift of data with missing views.
>
> **Q5.	Abstract claim that "Consensus representation imputation methods ignore the interview distribution bias due to missing views", what is the interview distribution bias? This concept may require more detailed visualizations to help understanding.**
>
> **R**: The inter-view distribution bias refers to the inherent distribution difference across views, which results in the inconsistency of consensus representation of incomplete multi-view data. Specifically, most incomplete multi-view clustering methods impute the missing views of a sample with its existing views and then fuse them into the consensus representation. Since there exist inter-view distribution bias, the consensus representation learned in this way will probably deviate from the ideal consensus representation learned from the complete views, which negatively impacts the clustering performance. We added the visualization results in the Figure 5 of the revised manuscript to help understanding.

---

### Meta-Review · Area_Chair_dkF1 · 2025-12-27

**Summary:**

The paper proposes PDD for incomplete multi-view clustering, using “unbiased projection refining biased projection” as the core idea: an attention-guided consensus projection refinement corrects distribution shift from missing views, while a denoising contrastive loss mitigates collapse under misalignment noise. The manuscript is clearly written and well organized, providing strong motivation, detailed methodology, and thorough analysis. Based on the recommendations of reviewers, I recommend acceptance for the paper.

**Reviewer Concerns:**

All reviewers gave positive comments, and concerns regarding the experiments and detailed descriptions were well addressed.

**Reviewer Scores:**

No explicit post-rebuttal upgrades and none negative scores.

---

### Decision · Program_Chairs · 2026-01-26

Accept (Poster)